# On The Influence of Rotary Dresser Geometry on Wear Evolution and Grinding Process

**DOI:** 10.3390/ma12233855

**Published:** 2019-11-22

**Authors:** Leire Godino, Jorge Alvarez, Arkaitz Muñoz, Iñigo Pombo

**Affiliations:** 1Faculty of Engineering Bilbao, University of the Basque Country (UPV/EHU), Plaza Ingeniero Torres Quevedo, 1, 48013 Bilbao, Bizkaia, Spain; arkaitzmu97@gmail.com (A.M.); inigo.pombo@ehu.eus (I.P.); 2Ideko S. Coop., Arriaga Kalea, 2, 20870 Elgoibar, Gipuzkoa, Spain; jalvarez@ideko.es

**Keywords:** OD grinding, CBN, dressing, rotary dresser, wear, CVD diamond

## Abstract

Dressing is a critical issue for optimizing the grinding process. Dresser tool and dresser parameters must be designed according to the grinding wheel material, shape, or even the dimensional and geometrical tolerances of the workpiece and its surface roughness. Likewise, one of the problematic issues of dressers is the wear that they suffer. In order to tackle this issue, the present work characterized the wear of two rotary dressers by analysing the wear behaviour depending on the pit radius of the dressers while studying the influence of the wear on ground surfaces. This work showed that the rotary dresser with a higher pit radius presents wear that is approximately 28% higher than the dresser with a half pit radius.

## 1. Introduction

The grinding process is a very important machining process characterized by the high added value of the ground parts. Its capacity for manufacturing advanced materials of poor machinability while achieving high-quality surfaces is made possible due to the combination of grinding wheel design, new advances in abrasives and an optimal dressing process. In recent years, the use of super-abrasive grinding wheels, primarily CBN (cubic boron nitride), has become increasingly extensive in the industry due to their thermal stability and the absence of chemical affinity with ferrous materials. Moreover, CBN grinding wheels are suitable when high removal rates are required and are used in creep feed grinding and high-speed grinding applications. Due to the different bond used with CBN grinding wheels, vitrified grinding wheels present more advantages than the other resin bond, metallic bond or electroplated grinding wheels. These advantages are primarily related to the ability to transport the coolant, the efficiency of chip removal and the capacity for dressing using rotatory dressing tools. Moreover, CBN grinding wheels are not the only widely used wheel in industrial applications, with conventional grinding wheels also being used due to their versatility and low cost in comparison with super-abrasives. The latest advances make them more competitive with regard to abrasive grain shape and crystalline structure.

One of the most important aspects of the grinding process is dressing, which is used in order to regenerate the abrasive capacity of the grinding wheel and the initial shape of the abrasive tool. Among the various types of dressers, these can be classified as static and rotary. The present work was focused on the rotary dressing tool, which consists of a cylindrical body with a single layer of inserted or deposited diamond particles. Different designs of rotary dressers can be found depending on the shape and size of the diamonds and their location, as well as their bonding chemical compositions. The main advantage of using this tool is the high dressing speed that can be achieved in comparison with a stationary single or multiple diamond tool, along with the possibility of generating a complex profile on the grinding wheel surface [1]. In addition, the wear level suffered by the dresser is relatively low [2]. Taking this fact into account, these types of dressers are used for dressing CBN and diamond grinding wheels with vitreous bond and conventional profile grinding wheels. The main advantages of rotary dressing tools are the high dressing speed and the possibility of generating complex profiles in the grinding wheel [1]. Moreover, this type of dressing tool provides the process with a high level of repeatability and precision, reducing processing costs and the number of rejected parts.

However, the complexity of the process is high, particularly when compared with the one carried out using a stationary diamond tool. In this regard, specific works focused on the influence of particular dressing parameters on the characteristics of the wheel surface can be found in the specialized literature. In [3], the first attempts were made to analyse the influence of dressing parameters on the wheel performance when using roller diamond tools. The Schmitt diagram is currently used in the industry to design dressing processes. In [2], a more in-depth analysis was conducted on the state of the art regarding grinding wheel conditioning, including the rotary dressing process. This work shows that a considerable amount of research has focused on the influence of the dressing parameters on grinding wheel performance, including the influence of the radial feed of the dressing process on the roughness of the wheel surface, theoretical pathways of dressing grains, influence of dressing speed ratios on the radial dressing force and effective grinding wheel roughness or the influence of the depth of dressing cut on specific normal grinding force. In addition, some studies on diamond tool wear were included, particularly [4] and the studies of Linke and Klocke [5,6], which focused on analysing diamond tool wear for stationary dressers.

Recent works regarding the analysis of the rotary dressing process are worth mentioning. For instance, in [7], the authors analysed the performance of various diamond tools under the same dressing conditions for small grinding wheels for internal grinding. In [8], a new type of rotary diamond tool was presented. In this case, the geometry and density of abrasive grits was completely monitored. These authors evaluated the performance of the new dresser against conventional dressers in terms of force and wear of the abrasives. The results were positive and the forces for the new dresser were found to be approximately 50% lower than that of the classic dressers, while the wear was also improved, although no quantitative values were reported. In [9], an analysis was presented regarding the performance of a roller dresser for the case of the micro-grinding of a titanium alloy. The authors focused their study on analysing the influence of the overlap ratio (U_d_) on the ground surface quality. The main conclusion reached by the authors was that extremely high values of overlap ratios were suitable for achieving high surface quality in micro-grinding processes. Finally, Palmer et al. [10] analysed the characteristics of the dressed wheel surface when using roller dressers.

Although one of the most relevant issues in dressing is the wear of the dresser, very little information can be found in the specialized literature regarding the wear of rotary dressers. This fact is very important when using this kind of dressing tool for two main reasons. The first is related to the characteristics of the dressed surface. In the case of rotary dressing tools, excessive wear of the dressing tool implies changes in the tool geometry and hence the tool sharpness parameter. This could imply changes in a_d_, or even in U_d_, and in the characteristics of the dressed surface. In Figure 1, the influence of a worn dressing tool on a plane grinding wheel is shown, with a smaller a_d_ being achieved for the worn rotary dresser.

The second reason is related to the geometry of the dressed wheel for non-plane profiles in the wheel. Depending on the geometry of the dressing tool, a dressing path is programmed in the machine. Once the rotary diamond tool has lost its shape due to wear after several dressing passes, either the dressing path or the rotary tool must be changed. If the wear suffered by the rotary diamond tool is not controlled, several ground parts could be rejected. Figure 2 shows the effect of grinding a profile wheel with a worn rotary dressing tool. In the left part of the image, the theoretical wheel profile is shown in green, and also points to the programmed path of the rotary dresser. In contrast, in the right part of the image, the path of the worn rotary dressing tool is shown. In this case, the real wheel profile is going to be larger than theoretical profile. Thus, when the part is ground, of the removed material is higher than programmed ones. This effect leads to rejection of the part.

In order to tackle these problems, the present work constitutes a preliminary approach toward characterizing the wear of rotary dressers. From research point of view, there is not a unique parameter to define the wear of rotary dressers. In contrast, the volumetric wear of grinding wheels is defined by G-ratio and the wear of stationary dressers is quantified using dressing wear ratio, G_d,_ applied by Shi and Akemon [11] for stationary blade diamond tools. Therefore, the aim of the present work was to define a wear parameter to quantify the wear suffered by the rotary dressing tool. A new parameter, termed “wear parameter, W_d_”, was presented. This parameter allowed the characterization of rotary dresser wear in order to be comparable to the stationary blade dressers. To this end, one of the objectives was to develop a systematic methodology for analysing and characterizing the wear suffered by a rotary dressing tool. The proposed methodology included the development of specific software (in Python) to measure the rotary diamond tool wear, the proposal of a parameter to measure such wear and an analysis of the grinding wheel behaviour, paying particular attention to the consumed power and surface roughness. This methodology was used to analyse the wear suffered by different geometry rotary dressing tools when dressing plane profile CBN grinding wheels with a vitreous bond.

For this purpose, the employed experimental setup and methodology was first presented. Second, the results are analysed and the W_d_ is defined. Finally, the main conclusions drawn from this work is presented.

## 2. Materials and Methods 

This study examined the wear of rotary dressing tools and its influence on ground workpieces. To this end, dressing and grinding tests were combined on a cylindrical grinding machine (DanobatGroup, Elgoibar, Spain). The study consisted of the analysis of two rotary dressers, varying the diamond pattern and the dresser geometry, i.e., the tip radius. RIG 52035 and RIG 52034 were manufactured by TYROLIT. For simplicity, the two rotary dressers are referred to as RIG 35 and RIG 34, respectively. Table 1 lists the main characteristics of the rotary dressers. In both cases, CVD (cultivated diamond) diamonds were inserted. In the case of RIG35, the interlayer was positioned. In contrast, in RIG34 dresser, the CVD diamonds were aligned. Regarding to the geometry of the rotary dressers, RIG35 presented a pit radius of 0.5 mm, whereas the pit radius of RIG34 was 0.25 mm. As previously mentioned, rotary dresser geometry is essential when profile grinding wheels are dressed. Any variation in the rotary dresser geometry is copied on the wheel surface.

In the present work, a CBN grinding wheel was used to conduct dressing and grinding tests. In order to distinguish between the influence of the wear on the ground workpiece surface and the grinding wheel shape, a straight grinding wheel was used. Thus, the influence of dresser wear on the wheel shape was not taken into account in this analysis, and only ground surface quality was analysed. The nomenclature of the used grinding wheel was CBN170N100V (UNESA S.L, Hernani, Spain), which was a medium hard wheel, presenting a vitreous bond and high density of abrasive grains. The external diameter of the wheel was Ø450 mm, the width was 10.3 mm, and the grain size was approximately Ø170 µm, corresponding to a finishing grinding wheel. Furthermore, plunge grinding tests were carried out on a cylindrical workpiece of hardened steel (AISI 52100). The medium hardness of the workpiece was approximately 54 HRC, while the external diameter of the parts was Ø80 mm.

The present study involved dressing and grinding, the analysis of the wear suffered by the rotary dresser, and the influence of wear on the grinding wheel surface and hence on ground workpiece surfaces. Accumulative dressing tests and plunge grinding tests were conducted using the same cylindrical grinding machine, *DANOBAT FG600S* (© DanobatGroup, Elgoibar, Spain), as shown in Figure 3. Moreover, CBN grinding wheel was used at a cutting speed of 50 m/s, and a water-based coolant with a concentration of 3.2% was used at a pressure of 13 bar. Furthermore, to conduct a complete analysis of the process during both dressing and grinding, real power consumption was measured using a *Load Control UPC* (© Load Controls Incorporated, Sturbridge, MA, USA) power meter, and a *USB-6008* data acquisition card from *National Instruments*. Additionally, in order to quantify the influence of dresser geometry on wear evolution and the effect of wear on the subsequent grinding process, a new methodology was developed, which is described in the following section. The complete approach was validated through experimental grinding tests under industrial grinding conditions, as detailed below.

Table 2 lists the dressing and grinding parameters. Moreover, the two dresser tools presented a similar dressing overlap ratio in order to compare the influence of the pit radius in the wear of the rotary dresser and in the ground surface. The overlap ratio is the relation between the effective width of the dressing tool and the feed per wheel revolution (U_d_ = b_d_/f_d_). A high value of dressing overlap ratio leads to a smooth grinding wheel surface, but the grinding forces and the specific energy are high. In contrast, a low value of dressing overlap ratio generates a sharper surface, with more cutting edges, thereby decreasing grinding forces. The range of values for the dressing process is U_d_ = 2–20 [12]. For the present study, a smooth wheel surface was required. Thus, the dressing overlap ratio was U_d_ = 2.64 for RIG 35 and U_d_ = 2.017 for RIG 34. Additionally, the smoothness of the grinding wheel surface depends not only on the dressing overlap ratio, but also on the dressing sharpness ratio, which is defined as the relation between the dressing depth and the effective width of the dresser (γ_d_ = a_d_/b_d_). This parameter represents the influence of the dresser shape on the wheel surface. In the present study, in both cases rotary dressers were studied, varying the pit radius. Thus, γ_d_ = 0.05 for RIG 35 and γ_d_ = 0.07 for RIG 34.

Accumulative dressing tests were carried out for each dressing tool, removing a total of 49,415 mm^3^ of wheel volume. The first step was to measure the new surface of the rotary dresser. The topography of new rotary dresser was characterized using a Confocal microscope Leica DCM3D^®^ (Leica microsystems AG, Wetzlar, Germany). Once the starting surface was characterized, the dressing test was carried out. Moreover, in Table 2, fine-dressing parameters are built. The CBN grinding wheel was continuously dressed, removing 12,354 mm^3^ of abrasive material. Immediately after the dressing process, the plunge grinding test was carried out. The specific rate of material removal during the grinding process was Q’_w_ = 5 mm^3^/mm s. After both the dressing and grinding tests, the worn surface of the rotary dressing tool and the ground surface were analysed. First, the dresser topography was measured again using a confocal microscope. The measurement after each dressing tests allowed for analysing the evolution of the wear during a complete test. Second, in order to observe the effect of dressing on the ground workpiece, the roughness of the ground surface is measured using a portable surface roughness tester (Taylor Hobson, Leicester, United Kingdom).

This step was completed a total of four times, following the same methodology, with 12,354 mm^3^ of abrasive material being removed at each step. Table 3 details the range of workpiece material removed at each step. The complete test was finished after dressing a total of 49,415 mm^3^. Similarly, during both dressing and grinding tests, the power consumption was measured in order to analyse the influence of dressing with a worn rotary dressing tool on the efficiency of the process. The last step involved the analysis of the data obtained during the tests. Power was readily analysed, while the topography data had to be processed and analysed to characterize the wear, which was a complex process. Thus, in the present work, a methodology for quantifying dresser wear was proposed, which is detailed hereafter.

### A Methodology for Quantifying Wear in Rotary Dressers

As a first step, it was necessary to accurately establish the geometry of the brand-new rotary dresser. To this end, specific tooling was designed for the dresser to take measurements on a Leica DCM3D^®^ confocal microscope (Leica microsystems AG, Wetzlar, Germany), as shown in Figure 4. The tooling system, together with the first measurement of the dresser surface, were necessary to set the references required to quantify the wear parameters.

Using this equipment, dresser topographies were obtained. Each state of wear was measured in four different zones along the profile of the disk, separated by 90 degrees. This is shown in the first image of Figure 5. The complete measured area was 2.546 × 8.477 mm^2^ and 2.808 mm in height, with a height resolution of 12 µm. The blue light was used in order to avoid dresser surface brightness. Three-dimensional (3D) profiles were then extracted, as seen in the second image. Profile comparison was carried out by slicing the 3D geometry and obtaining 2D curves. Five curves were obtained for each 3D profile. For this purpose, the topography layer of the LeicaMap^®^ (Leica microsystems AG, Wetzlar, Germany) was used, as shown in the third and fourth images of Figure 5. At this stage, it is important to note that the reference must be set at the diamond and not at the bonding, since the latter will suffer more pronounced wear. Therefore, the intensity layer (shown in the third image of Figure 5) of the data was used as a reference because, on this layer, the infiltrated diamonds can be clearly observed and slices were made to coincide with CVD diamonds. Thus, only the wear of the diamond was taken into account, avoiding the influence of the bond. Moreover, this layer used the same scale as that used by the topography layer from which the 2D profiles were extracted. It is important to note that the rotary dresser profiles could not be obtained using a stylus profilometer due to the shape of the dresser surface and the abrasive surface. Therefore, a confocal microscope is the best option to analyse this kind of surface.

Once the 2D profiles are available, it is possible to compare different states of wear of the dresser. Various geometrical parameters can be selected for comparison. In this work, two auxiliary parameters (namely worn area and contact length) and one main wear indicator (wear height, h_d_) were defined. To obtain these parameters, profiles at different wear stages must be overlapped while maintaining stable references. In order to do so, a Python app was developed.

The first utility developed to assist profile comparison was used for profile smoothing. Any possible measurement defect, or even the presence of noise on the signal, was filtered using a low band pass filter. After some experimentation, the order of the filter was set at 6, the sample frequency at 15 Hz, and the cut-off frequency at 1.5 Hz. The second step involved overlapping the profiles at different wear stages. Errors were removed using a best-fitting technique on the profiles. In order to apply the best-fitting technique, reference points were set. These points did not suffer wear during dressing because they were not in contact with the grinding wheel. In Figure 6a, it is shown that points 1 and 2 are the reference points of corresponding new and worn profiles. Moreover, in Figure 6b, the two overlapping profiles show the wear suffered by the rotary dresser tool.

The contact area was defined to correspond to the limits set by the points where deviations were more important in value. Subsequently, the worn area and the contact zone can be effectively quantified, as shown in Figure 7a. The worn area can later be used to estimate the total volume of dresser worn during the operation. Further, maximum and average values of the wear height parameter h_d_ can be obtained at this stage, as shown in Figure 7b.

## 3. Results and Discussion

In this section, the wear of the rotary dressers with CVD infiltrated diamonds was analysed. First, the influence of dresser geometry on the wear was studied, after which the influence of dresser wear on the surface quality of the workpiece after dressing was assessed. The majority of works that have analysed dresser wear used the wear volume (V_d_), the wear of removed grinding wheel (V_w_), and thus, the dressing wear ratio (G_d_), in order to quantify the wear of the dressers, to compare different wear states, and to determine dresser life. Almost all the studies have been conducted for stationary dressers, namely single point or blade diamond tools [11]. In contrast, in the present work, dressers of differing geometry were analysed, and these parameters were not appropriate for making the comparison. Therefore, wear height was used due to the dissimilarities in pit radius and hence the wear volume of both the studied rotary dressers, as Figure 8a shows. For a given h_d_, V_d_ can be approximately half the value for RIG 34 in comparison with RIG 35. Moreover, the height is the parameter that affects dressing precision and, as a consequence, the quality of the grinding process. If the rotary dresser decreases in diameter due to the height wear, the wheel does not achieve the designed dimensions and shape, obtaining larger dimensions. The error is translated to the workpiece during grinding, as removing more than the corresponding material would cause the workpiece to be rejected. Thus, it was necessary to analyse the evolution of h_d_ during the process.

Figure 8b represents the evolution of an accumulative dresser wear with the workpiece removed from the material. A linear increase in wear was shown with V_w_ for both rotary dressers. RIG35 presented a slightly higher slope than RIG34, achieving higher wear at the end of the tests. When 49,415 mm^3^ of grinding wheel was removed, the rotary dresser RIG 35 presented a level of wear that was 28% higher than RIG 34. Thus, RIG 35, which had a greater pit radius, showed a higher tendency towardswear than RIG 34. In this regard, it is necessary to highlight that if the comparison had been made through V_d_, the wear difference between RIG 34 and RIG 35 would have been 145%, and also higher for RIG35. This result indicates that the volumetric parameters are valid for comparing dressers with the same geometry when the compared volume is equivalent. Thus, the parameter for determining the wear of two rotary dressers that differ in geometry—in this case, different pit radius or even diameter—must be the wear height.

Moreover, dressing volumetric ratio, i.e., the relation between the removed volume of the wheel and the worn volume of the rotary dressing tool (G_d_ = V_w_/V_d_), is also used to determine dressing process efficiency in the case of stationary dressers. However, as mentioned, this is not a valid parameter for comparing rotary dressers of differing geometry. In this regard, the present work measured efficiency through the relation between removed volume of the wheel and the height of the wear of the rotary dressing tool, defining the dressing wear parameter (W_d_ = V_w_/h_d_). High values of W_d_ imply that the rotary dresser suffers less wear when dressing a higher quantity of wheel volume.

Regarding the case studied here, in Figure 9, the rotary dresser with lower pit radius, RIG34, presents a higher value of W_d_ than RIG35. Thus, when using a rotary dresser with a half pit radius, the W_d_ was approximately 31% higher in the case under study. Thus, RIG 34 was more efficient than RIG 35, increasing dresser life and hence becoming a more economic process due to the high cost of CVD rotary dressers. With the wear values obtained, both h_d_ and W_d_, it can be confirmed that the rotary dresser with smaller pit radius presented less wear in height, as well as higher dressing efficiency for the studied dressing and grinding parameters.

Figure 10 plots the mean value of power consumption during grinding after each dressing step. To analyse the power, only the three last steps of the test, from 24,707 mm^3^ until the end of the test, were taken into account. Thus, the first contact, in which the dresser wear presented a transitory behaviour, was avoided. Thus, from 24,707 mm^3^ to the end of the test, power consumption during grinding decreased for both rotary dressers. However, when comparing the two rotary dressers, power consumption tended to decrease more markedly after dressing with RIG35 in comparison with RIG34. If the grinding wheel was dressed using the RIG35, the power consumption during grinding was approximately 18% higher than if the wheel was dressed using the RIG34 rotary dresser. This could be due to the influence of the loss of sharpness ratio on the wheel surface.

The initial sharpness ratio was γ_d_ = 0.05 for RIG35 and γ_d_ = 0.07 for RIG34. Thus, RIG 34 was sharper, leading to a smoother wheel surface with more active (but shallower) cutting edges, as shown in the upper part of Figure 11a. When dressers were worn, the sharpness ratio decreased. The resulting wheel surface had fewer (and deeper) cutting edges than at the beginning of the tests. This helps to remove material during grinding, consuming less power. However, this is not the case for any dresser wear state, that is, it only occurs if the dresser is slightly worn. Analysing Figure 9a and taking into account the shape of rotary dressers, for the case studied here, RIG34 presented lower power consumption during grinding because the dressing process generated a greater quantity of (more shallow) cutting edges. Likewise, for RIG 35, in the last state it can be observed that the power consumption was similar. From this point, the power increased because the wear of the dresser was too high.

Finally, the influence of dressing on the quality of the workpiece surface was analysed. To this end, the roughness of ground workpieces was measured. In Figure 11c, the real profile generated by the rotary dresser tool RIG 35 is shown. During a complete test, higher Ra was achieved for RIG 34 than for RIG35. Thus, RIG35 led to smoother ground surfaces with the used parameters. If the evolution of the roughness during the test is analysed, different behaviour is shown in both cases, as displayed in Figure 11b. For RIG35, first, slightly lower values of Ra were measured (approximately 0.26 µm), and 0.3 µm were observed by the end of the test. In contrast, RIG34 did not present a tendency toward roughness, and the values varied from 0.35 to 0.4. In the last studied state, the difference in roughness between the two surfaces was lower than 16%. In any case, the values of Ra obtained were lower than 0.4 µm. Thus, a good surface quality was achieved despite the wear of rotary dressers.

## 4. Conclusions

After conducting a complete analysis of worn dressers and studying the influence of worn dressers on the ground workpiece surface, the obtained results were carefully analysed. From a discussion of the results, the following conclusions can be drawn:It was demonstrated that dressing wear ratio, G_d_, is not an adequate parameter to quantify the wear of rotary dressers. If wear volumes are compared, the results indicate that RIG35 presented wear approximately 145% higher than RIG34. In contrast, if heights, h_d_, are compared, the difference is around 28%.Dressing wear parameter (W_d_ = V_w_/h_d_) was defined in order to compare rotary dressers of different pit radius and diamond density.With regard to the height of the wear of rotary dressers, h_d_, a linear increase of was shown during a complete test. Moreover, the dresser with a higher pit radius presented a wear about 60 µm higher than that with a half pit radius.Regarding to W_d_, results demonstrated a more efficient dressing process for lower pit radius. Thus, in the present study, when using a rotary dresser with a half pit radius, the W_d_ was approximately 31% higher.Regarding the grinding process, the rotary dresser with the higher pit radius presented relatively lower power consumption. Dressing with higher pit radius led to a wheel surface with fewer, albeit deeper, cutting edges. Thus, the material was removed easily from the workpiece, consuming less power.Finally, grinding after dressing with both pit radius led to good surface quality, with an Ra lower than 0.4 µm.

These results suggest that for the range of dresser life studied here, the wear suffered was around 60 µm in height for RIG35 and 47 µm for RIG34, while the roughness of the ground workpiece did not undergo any significant changes. Moreover, it is necessary to bear in mind that this analysis was carried out for a plane grinding wheel, so the influence of the geometry lost in the wheel profile was not analysed. In this preliminary approach, only the plane grinding wheels were tested. Thus, it will be of interest to tackle the problem of non-plane grinding wheels in future works.

## Figures and Tables

**Figure 1 materials-12-03855-f001:**
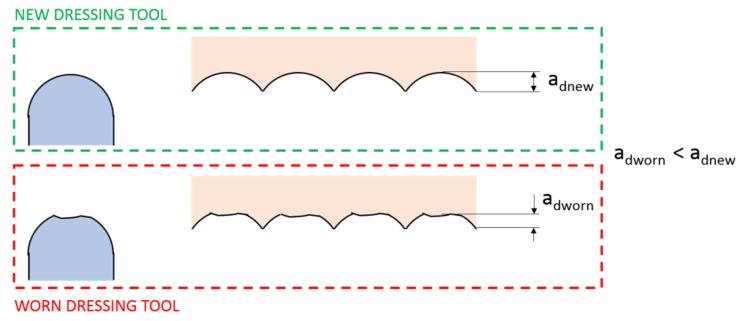
Influence of the worn rotary dressing tool on the wheel surface.

**Figure 2 materials-12-03855-f002:**
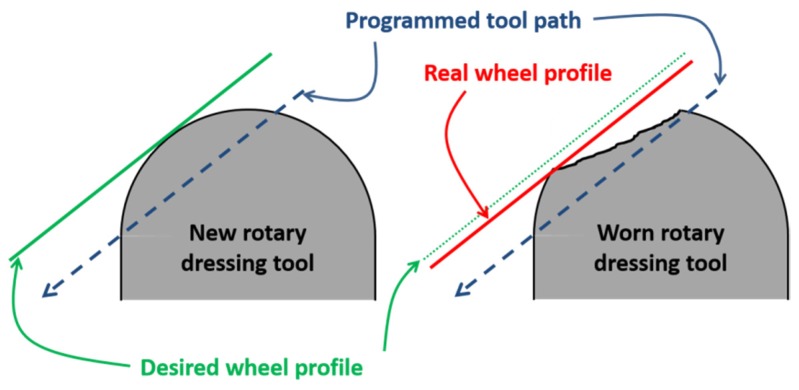
Influence of the worn rotary dressing tool on the wheel profile.

**Figure 3 materials-12-03855-f003:**
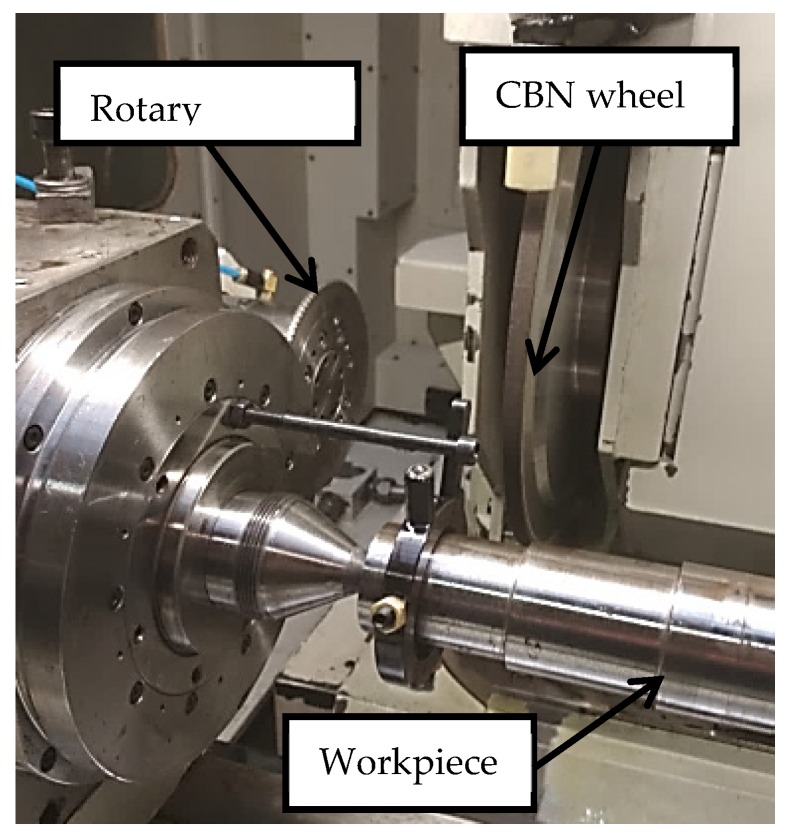
Dressing and grinding test set up.

**Figure 4 materials-12-03855-f004:**
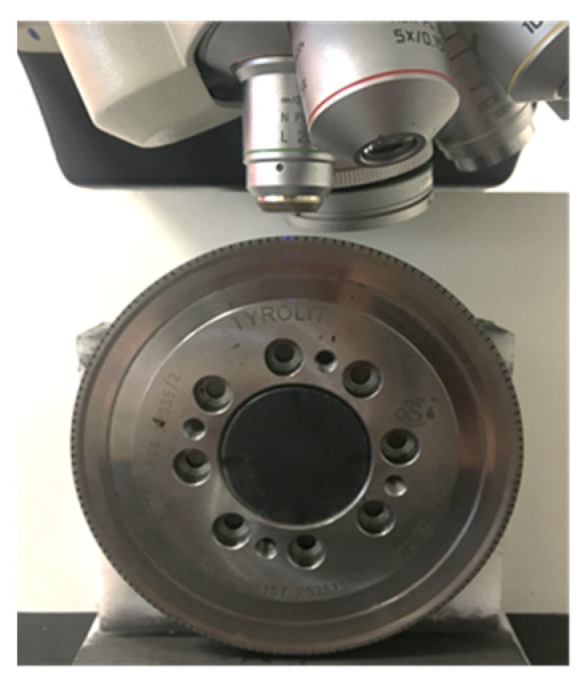
Initial characterization of the geometry of the rotary dresser on the confocal microscope.

**Figure 5 materials-12-03855-f005:**
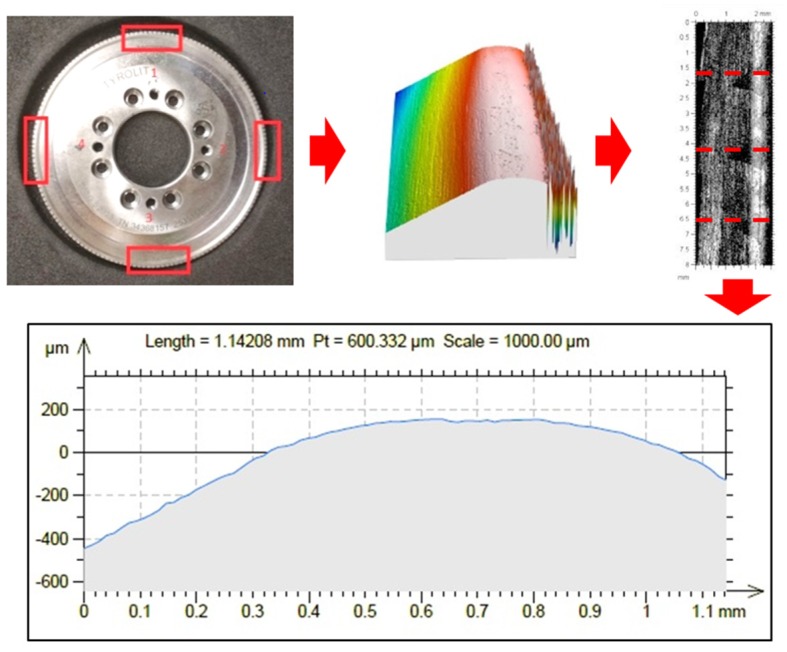
Rotary dresser measurement to characterize the wear.

**Figure 6 materials-12-03855-f006:**
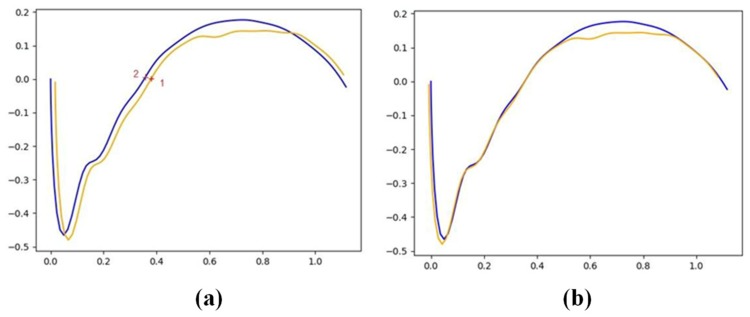
(**a**) Incorrect overlapping of profiles 1 and 2; (**b**) solution after applying best-fitting technique.

**Figure 7 materials-12-03855-f007:**
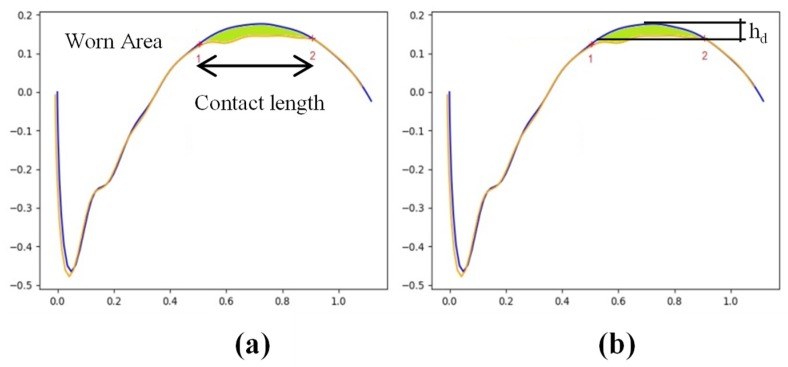
(**a**) Contact area and zone of the dresser where wear concentrates; (**b**) definition of wear height h_d_.

**Figure 8 materials-12-03855-f008:**
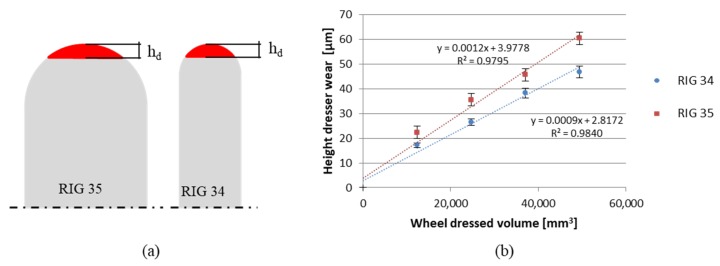
(**a**) Dresser comparison and (**b**) wear height evolution during a complete test.

**Figure 9 materials-12-03855-f009:**
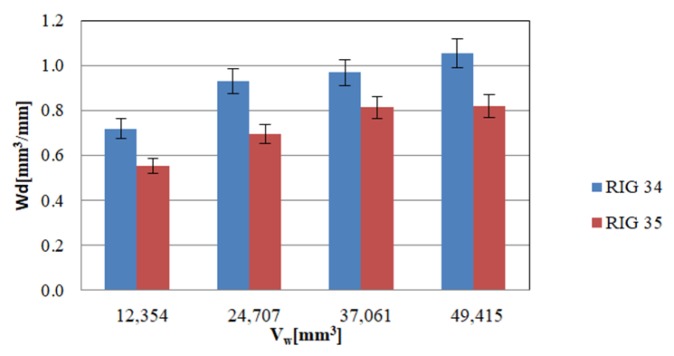
Dressing wear parameters.

**Figure 10 materials-12-03855-f010:**
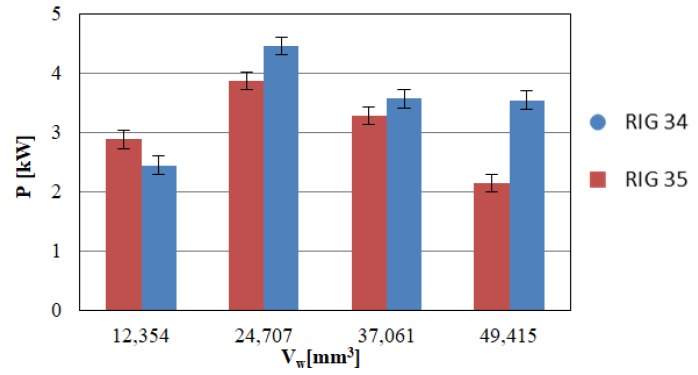
Power consumption during grinding.

**Figure 11 materials-12-03855-f011:**
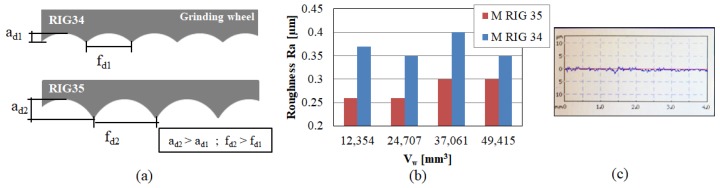
(**a**) Influence of sharpness ratio and the shape of the rotary dresser on the wheel surface, (**b**) roughness of workpiece after grinding and (**c**) the real Ra profile corresponding to 0.26 µm.

**Table 1 materials-12-03855-t001:** Characteristics of rotary dressing tools.

	RIG 35	RIG 34	Common Characteristics	
Diamond pattern	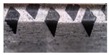	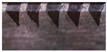	Diamond type	CVD
			External diameter [mm]	120
Pit radius [mm]	0.5	0.25	Internal diameter [mm]	40
Width [mm]	4.26	3.88	Internal angle	35°

**Table 2 materials-12-03855-t002:** Dressing and grinding parameters.

Dressing Parameters		Grinding Parameters	
Cutting speed v_s_ [m/s]	50	Cutting speed v_s_ [m/s]	50
Dresser speed n_d_ [rpm]	3453	Feed rate v_f_ [mm/min]	1.194
Dressing depth a_d_ [µm]	10	Depth of cut a_e_ [µm/rev]	8
Speed ratio between dresser and wheel	0.434	Speed ratio between wheel and workpiece	80
Dressing feed rate v_fd_ [mm/min]	100	Removed material in diameter [µm]	300

**Table 3 materials-12-03855-t003:** Range of material removed from the workpiece for each step of a given test.

Complete Test			
Step 1	Step 2	Step 3	Step 4
0-12354 mm^3^	12,354–24,707 mm^3^	24,707–37,061 mm^3^	37,061–49,415 mm^3^

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
