# Peer review of "On The Influence of Rotary Dresser Geometry on Wear Evolution and Grinding Process"

_materials, 2019, doi:10.3390/ma12233855_

Round 1

Reviewer 1 Report

144-146 Authors write that grinding using CBN grinding wheel needs high-pressure coolant, but in the study there was pressure of only 13 bar used. Please explain this.

174-182 Authors repeat grinding parameters from table 2.

Fig. 5 Figure 5 has poor quality.

Fig. 6 Figure 6 is redundant. Please describe algorithm of best fitting technique instead of figure 6.

297-299 Text does not correspond with figure 9

314-323 Changes in surface roughness arise from dresser geometry and are obvious

Main conclusion: Authors stated that difference between dressers wear arises from pit radius differences. Authors don’t take into account diamond pattern difference between RIG 35 and RIG 34. RIG 34 has more dense diamond distribution so its wear is less than RIG 35. Please explain this.

Conclusions should be more compact.

Author Response

First of all, the authors want to express their gratitude to the reviewer for his/her effort and time in analysing the manuscript. His/her criticisms and suggestions have been really valuable to improve the quality of the manuscript.

144-146 Authors write that grinding using CBN grinding wheel needs high-pressure coolant, but in the study there was pressure of only 13 bar used. Please explain this.

The sentence was confused, so I have rewritten it:

Accumulative dressing tests and plunge grinding tests were conducted using the same cylindrical grinding machine, DANOBAT FG600S, as shown in Figure 3. Moreover, CBN grinding wheel is used at a cutting speed of 50m/s and a water-based coolant with a concentration of %3.2 was used at a pressure of 13 bar.

174-182 Authors repeat grinding parameters from table 2.

The paragraph after Table 2 is summarized, avoiding repeating grinding and dressing parameter:

Accumulative dressing tests were carried out for each dressing tool, removing a total of 49415mm3 of wheel volume. The first step was to measure the new surface of the rotary dresser. The topography of new rotary dresser is characterized using a confocal microscope Leica DCM. Once the starting surface is characterized, the dressing test can be carried out. Moreover, in Table 2 fine-dressing parameters are built. The CBN grinding wheel was continuously dressed, removing 12354 mm3 of abrasive material. Immediately after the dressing process, the plunge grinding test was carried out, being the specific rate of material removal during the grinding process Q’w=5 mm3/mm s. After both the dressing and grinding tests, the worn surface of the rotary dressing tool and the ground surface were analysed. Firstly, the dresser topography was measured again using a confocal microscope. The measurement after each dressing tests allows for analysing the evolution of the wear during a complete test. Secondly, in order to observe the effect of dressing on the ground workpiece, the roughness of the ground surface is measured using a portable surface roughness tester (Taylor Hobson).

5 Figure 5 has poor quality.

The quality of the image is improved.

6 Figure 6 is redundant. Please describe algorithm of best fitting technique instead of figure 6.

The description is improved

Errors are removed by using a best-fitting technique on the profiles. In order to apply best-fitting technique, a reference points are set. These points do not have suffered wear during dressing because they are not been in contact with the grinding wheel. In Figure 6 (a) it is shown that points 1 and 2 are the reference points of corresponding new and worn profiles. Moreover, in Figure 6 (b) the two profiles overlapped show the wear suffered by the rotary dresser tool.

297-299 Text does not correspond with figure 9

The test order and the figures are changed.

314-323 Changes in surface roughness arise from dresser geometry and are obvious

We decided to add the roughness value in order to present a complete study of the rotary dresser tool wear. Likewise, in the conclusions, the point corresponding to the roughness is corrected.

Main conclusion: Authors stated that difference between dressers wear arises from pit radius differences. Authors don’t take into account diamond pattern difference between RIG 35 and RIG 34. RIG 34 has more dense diamond distribution so its wear is less than RIG 35. Please explain this. Conclusions should be more compact.

All the conclusions are rewritten as follow:

It is demonstrated that dressing wear ratio, Gd, is not an adequate parameter to quantify the wear of rotary dressers. If wear volumes are compared, the results indicate that RIG35 presents wear that is approximately 145% higher than RIG34. In contrast, if heights, hd, are compared, the difference is around 28%. Dressing wear parameter (Wd= Vw/hd) is defined in order to compare rotary dressers of different pit radius and diamond density. With regard to the height of the wear of rotary dressers, hd, a linear increase of is shown during a complete test. Moreover, the dresser with a higher pit radius and presents a wear about 60µm higher than that with a half pit radius. Regarding to Wd, results demonstrate a more efficient dressing process for lower pit radius. Thus, in the present study, when using a rotary dresser with a half pit radius, the Wd is approximately 31% higher. Regarding the grinding process, the rotary dresser with the higher pit radius presents relatively lower power consumption. Dressing with higher pit radius leads to a wheel surface with fewer, albeit deeper, cutting edges. Thus, the material is removed easily from the workpiece, consuming less power. Finally, grinding after dressing with both pit radius leads to good surface quality, with an Ra lower than 0.4 µm.

Reviewer 2 Report

Very interesting article, but the scientific field is not new. I think the authors should more clearly define the worth of their work.

Author Response

First of all, the authors want to express their gratitude to the reviewer for his/her effort and time in analysing the manuscript. His/her criticisms and suggestions have been really valuable to improve the quality of the manuscript.

Very interesting article, but the scientific field is not new. I think the authors should more clearly define the worth of their work.

In the two last paragraphs of the introduction is detailed the scientific advances of this paper:

In order to tackle these problems, the present work constitutes a preliminary approach towards characterizing the wear of rotary dressers. From research point of view, there is not a unique parameter to define the wear of rotary dressers. In contrast, the volumetric wear of grinding wheels is defined by G-ratio and the wear of stationary dressers is quantified using dressing wear ratio, Gd, applied by Shi and Akemon [11] for stationary blade diamond tools. Therefore, the aim of the present work is to define a wear parameter to quantify the wear suffered by the rotary dressing tool. A new parameter termed “wear parameter, Wd”, will be presented. This parameter allows the characterization of rotary dresser wear in order to be comparable to the stationary blade dressers. To this end, one of the objectives is to develop a systematic methodology for analysing and characterizing the wear suffered by a rotary dressing tool. The proposed methodology will include the development of specific software (in Python) to measure the rotary diamond tool wear, the proposal of a parameter to measure such wear, and an analysis of the grinding wheel behaviour, paying particular attention to the consumed power and surface roughness. This methodology will be used to analyse the wear suffered by different geometry rotary dressing tools when dressing plane profile CBN grinding wheels with a vitreous bond.

For this purpose, the employed experimental set up and methodology will first be presented. Secondly, the results will be analysed and Wd, will be defined. Finally, the main conclusions drawn from this work will be presented.

Reviewer 3 Report

The authors of this paper studied the effect of the geometry of the rotary dresser on wear evolution.

Page 3, please improve the last sentence before Figure 2.

Page 4, just before Table 1; there should be 54 HRC.

Table 1, is the external diameter correct?

Table 2 – please improve the unit of depth of cut.

Please add details about the measurement of surface topography using a confocal microscope and about the measurement of surface roughness using a stylus profilometer. Why have you used various measuring devices?

Please add details about overlapping of profiles (Figure 6).

Please add error bars to Figure 8b and Figure 9.

Please show roughness profiles of workpieces after grinding.

Author Response

First of all, the authors want to express their gratitude to the reviewer for his/her effort and time in analysing the manuscript. His/her criticisms and suggestions have been really valuable to improve the quality of the manuscript.

Page 3, please improve the last sentence before Figure 2.

The sentence is changed as follow:

In this case, the real wheel profile is going to be larger than theoretical profile; thus, when the part is ground, of the removed material is higher than programmed ones. This effect leads to rejection of the part.

Page 4, just before Table 1; there should be 54 HRC.

It is changed in the manuscript:

54 HRC

Table 1, is the external diameter correct?

Yes, the external diameter is correct

Table 2 – please improve the unit of depth of cut.

Units are corrected:

Depth of cut ae [µm/rev]

Please add details about the measurement of surface topography using a confocal microscope and about the measurement of surface roughness using a stylus profilometer. Why have you used various measuring devices?

I have add some measurement parameters for confocal. Moreover, I have explained why it is necessary to measure the dresser surface with a confocal microscope. However, the workpiece roughness can be measured using a stylus profilometer, with is more fast and more accurate because it is a contact method.

Using this equipment, dresser topographies are obtained. Each state of wear is measured in 4 different zones along the profile of the disk, separated by 90 degrees. This is shown in the first image of Figure 5. The complete measured area is 2.546 x8.477 mm2 and 2.808 mm in height, with a height resolution of 12 µm. The blue light is used in order to avoid dresser surface brightness. 3D profiles are then extracted, as can be seen in the second image. Profile comparison is carried out by slicing the 3D geometry and obtaining 2D curves, 5 curves for each 3D profile. For this purpose, the topography layer of the Leica map software is used, as shown in the third and fourth images of Figure 5. At this stage it is important to note that the reference must be set at the diamond and not at the bonding, since the latter will suffer more pronounced wear. Therefore, the intensity layer (shown in the third image of Figure 5) of the data is used as a reference, because on this layer the infiltrated diamonds can be clearly observed, and slices are made to coincide with CVD diamonds. Thus, only the wear of the diamond is taken into account, avoiding the influence of the bond. Moreover, this layer uses the same scale as that used by the topography layer from which the 2D profiles are extracted. It is important to note that the rotary dresser profiles cannot be obtained using a stylus profilometer due to the shape of the dresser surface, and also due to the abrasive surface. Therefore, a confocal microscope is the best option to analyze this kind of surfaces.

Please add details about overlapping of profiles (Figure 6).

The description is improved

Errors are removed by using a best-fitting technique on the profiles. In order to apply best-fitting technique, a reference points are set. These points do not have suffered wear during dressing because they are not been in contact with the grinding wheel. In Figure 6 (a) it is shown that points 1 and 2 are the reference points of corresponding new and worn profiles. Moreover, in Figure 6 (b) the two profiles overlapped show the wear suffered by the rotary dresser tool.

Please add error bars to Figure 8b and Figure 9.

The error bars are included

Please show roughness profiles of workpieces after grinding.

The profile corresponding to the workpiece after grinding is included in the manuscript in figure 10 and the text is modified

Figure 10 (a) Influence of sharpness ratio and the shape of the rotary dresser on the wheel surface, (b) roughness of workpiece after grinding and (c) the real Ra profile corresponding to 0.26µm.

Finally, the influence of dressing on the quality of the workpiece surface is analysed. To this end, the roughness of ground workpieces is measured. In Figure 10 (c), the real profile generated by the rotary dresser tool RIG 35 is shown. During a complete test, higher Ra is achieved for RIG 34 than for RIG35, and thus, RIG35 leads to smoother ground surfaces with the used parameters. If the evolution of the roughness during the test is analysed, different behaviour is shown in both cases, as displayed in Figure 10 (b). For RIG35, firstly, slightly lower values of Ra are measured (approximately 0.26 µm) observing values of 0.3 µm by the end of the test. In contrast, RIG34 does not present a tendency towards roughness and the values vary from 0.35 to 0.4. In the last studied state, the difference in roughness between the two surfaces is lower than 16%. In any case, the values of Ra obtained are lower than 0.4 µm; thus, a good surface quality is achieved despite the wear of rotary dressers.

Round 2

Reviewer 1 Report

I suggest to remove fig. 6. It contains obvious information. The wear suffered by the rotary dresser tool is shown on the fig 7. Fig. 6 is redundant.

Author Response

Sorry for the confusion. Now I have removed Figure 6